# A Panel Data Analysis on Sustainable Economic Growth in India, Brazil, and Romania

**Batrancea Ioan [1],\*, Rathnaswamy Malar Kumaran [1], Batrancea Larissa [2], Nichita Anca [3], Gaban Lucian [3], Fatacean Gheorghe [1], Tulai Horia [1], Bircea Ioan [4] and Rus Mircea-Iosif [5]**

[1] Faculty of Economics and Business Administration, Babes-Bolyai University, 58–60 Teodor Mihali Street, 400591 Cluj-Napoca, Romania; malarkumaran9@gmail.com (R.M.K.); gheorghe.fatacea@econ.ubbcluj.ro (F.G.); horea.tulai@econ.ubbcluj.ro (T.H.)

[2] Faculty of Business, Babes-Bolyai University, 7 Horea Street, 400174 Cluj-Napoca, Romania; lm.batrancea@tbs.ubbcluj.ro

[3] Faculty of Economic Sciences, "1 Decembrie 1918" University of Alba Iulia, 15–17 Unirii Street, 510009 Alba Iulia, Romania; ancaramona.nichita@gmail.com (N.A.); lucian.gaban@uab.ro (G.L.)

[4] Faculty of Economics and Law, University of Medicine, Pharmacy, Science and Technology of Targu Mures, 38 Gheorghe Marinescu Street, 540142 Targu Mures, Romania; ioan.bircea@umfst.ro

[5] National Institute for Research and Development in Constructions, Urbanism and Sustainable Spatial Development "URBAN INCERC", 117 Calea Floresti, 400524 Cluj-Napoca, Romania; mircearus2005@yahoo.com

\* Correspondence: ioan.batrancea@econ.ubbcluj.ro; Tel.: +40-754077019; Fax: +40-264412570

**Abstract:** The study investigated the impact of factors such as non-performing loans, $CO_2$ emissions, bank credit, and inflation on the variable sustainable economic growth for India, Brazil, and Romania during the period 2005–2017, through a panel data analysis. Specifically, we investigated the timeline before, during, and after economic turmoil, with a special focus on the global financial crisis. Our empirical results are valuable for both developing and developed nations. As a first result, we showed that $CO_2$ emissions increased the level of economic growth, but in this context, authorities should design suitable policies to limit its impact on the overall society. In addition, a single supervision mechanism increased the level of sustainable economic growth. Last but not the least, the period during and after the global financial crisis, sustainable economic growth decreased under the influence of bank credit, inflation, and non-performing loans. Within this framework, public authorities are called to design efficient economic, fiscal, and monetary policies.

**Keywords:** $CO_2$ emissions; gross domestic product; sustainable economic growth; single supervision mechanism; global financial crisis; inflation; bank credit; non-performing loans

**JEL Classification:** E5; E6; GO1; G2; O1; Q01

## 1. Introduction

Considering the rapid population growth and the finite nature of non-renewable resources, nowadays, research studies focused on the topic of sustainable economic growth have become more and more important for academics (Batrancea et al. 2020), practitioners, international and regional organizations, national authorities, and citizens alike. Moreover, the global interconnection of markets demands that national authorities design and implement sound macroeconomic policies, in order to secure a sustainable economic growth in the long run. In the present research endeavor, we have focused on a country sample comprising India, Brazil, and Romania, in order to study factors that influence sustainable economic growth. The countries considered experience different levels of

economic development, political instability, rule of law, social injustice, economic disparity, and human development. Moreover, since they depend on agriculture and natural resources, these countries are also fast-growing economies. India, for instance, has the second largest population in the world and Brazil has the largest area of land, both countries being members of the BRICS group (i.e., acronym for the country association comprising Brazil, Russia, India, China, and South Africa). Romania is a fast-growing economy among the newly admitted members of the European Union (EU). India and Brazil follow single supervisory mechanisms for the banking sector, while Romania follows a dual supervisory mechanism. In the following paragraphs, we present various details about the economic realities of these countries.

Patil and Kadam (2014) find that the economic growth of India is not sustainable. According to a World Values Survey (2014) study covering 80 countries, 50% of respondents favored economic growth despite environmental degradation (Turaga 2016). Hence, there seems to be a conflict between environment and development (Bertelmus 2013; Rathnaswamy 2000; Samans 2015). The natural capital of eleven Indian states has declined from 2005 to 2015, while there has been consistent economic growth (Government of India 2018; Pandey 2018). The GDP growth rate in India increased to 8% in 2018, after a sluggish evolution due to demonetization and the introduction of the Goods and Services Tax (GST). According to the World Bank, India should pursue structural reforms to revive bank credit, strengthen its competitive environment, increase private investment, and accelerate green investments (World Bank 2018). India contributes 15% to global growth, therefore, its economy is considered to be consistently growing at a fast pace. Nevertheless, the credit available for corporate sectors and the profitability of banks have declined during the last years. The country has introduced a new insolvency and bankruptcy code to reduce non-performing loans and it has implemented reforms to improve the efficiency of public sector banks. India has numerous public banks (IMF 2018) and it has committed to implement IMF advice on macroeconomic policies.

Brazil has achieved a significant progress in reducing deforestation and emissions of greenhouse gases over the past 15 years. Considering the multitude of social and economic opportunities offered by green markets, Brazil could achieve up to 7% of its GDP in this direction. The country initiated various institutional changes that promote sustainable growth cycles, while taking advantage of the challenges imposed by recession and its negative GDP. Moreover, Brazil aims to increase its renewable energy with 45% of the total energy consumption, by the year 2030. On integrating principles of sustainable development into environmental policies, Brazil aims to protect biodiversity, improve the provision of clean water by 50%, eradicate poverty of 100 million slum dwellers, preserve freshwater, reduce child mortality, and regulate economic exploitation of its ecosystems. Therefore, green investments in Brazil would accelerate the transition of this agrarian society to an industrial society. Brazil achieved higher economy growth rates during the period 2003–2010, but this declined drastically from 2011 onwards, due to the impact of the global financial crisis from 2008–2009, more precisely.

According to the international rating agency, Moody's, the level of economic growth in Romania is not sustainable, as its GDP was forecasted to decline to 4% in 2018 and to 3.5% in 2019, because of negative exports, fiscal expansion and budget rigidity, political instability, weak rule of law, inadequate structural reforms, and a reduced fiscal buffer, which is insufficient to meet the challenges of potential shocks that hinder sustainable economic growth. Romania suffers from an unsustainable exploitation of natural resources and loss of biodiversity. Although a member of the European Union, the country maintains its own currency (i.e., "leu"), but is engaged in the pursuit of structural reforms. Deposits at domestic banks have increased their values, and consequently, the dependence of foreign banks declined drastically, despite the fact that in Romania there are 29 foreign banks out of the total of 35 banks. Generally, banks dominate the financial sector in Romania. The lending practice of the non-banking financial sector is likely to increase the non-performing loans.

Sustainable economic growth is one of the principles of sustainable development, as referred to in the Millennium Development Goals. Sustainable economic growth does not represent the end of sustainable development. It includes certain core principles of sustainable ecosystems, sustainable

consumption of renewable and non-renewable resources without depriving society of future benefits, sustainable human development, sustainable investment, and innovation. Sustainable development means achieving development without environmental degradation, which is self-contradictory in its core, since nature is one of the production factors (World Commission on Environment and Development 1987). Consequently, the exploitation of the natural capital would result in environmental degradation. In this context, sustainable economic growth suggests a transformation of the brown economy into a green or low-carbon economy. Thus, $CO_2$ emission would be regulated to mitigate global warming. Pearce et al. (1989) hold that sustainable economic growth ensures the per capita of human well-being, without fluctuation over time. Arrow et al. (2012) believe that sustainable development stops as economic growth declines. There is a link between environment and development. Green investments should be promoted, hence financial and banking sector play important roles in decreasing $CO_2$ emissions, while achieving sustainable economic growth. Augmenting crediting levels for green investments increases the money supply, which in turns causes inflation, and any repayment failures become non-performing assets or non-performing loans (NPL). On the other hand, higher investments lead to economic growth and a higher growth rate of the gross domestic product (GDP). According to Victor (2010, 2019), there is no need for economic growth, because it entails the exploitation of natural resources and, therefore, economic growth represents an option and not a compulsion. Furthermore, the author gives three arguments against the economic model. The first argument is that economic growth is not related to happiness. The second argument is that economic growth has widened inequality rather than reduced poverty levels. The third argument is that developed nations benefit more than developing nations from such type of economic growth. Consequently, Victor pleads for limiting growth and recommends the conventional model of Canada, which includes factors such as employment, the reduction of greenhouse gas emissions, and managing government debt.

Within the banking system, central banks are the supervisors of commercial banks. There are two well-known banking supervisory mechanisms—the single supervisory mechanism and the dual/multiple supervisory mechanism. A case in point, the European Union has a single supervisory mechanism in the Euro Area and a multiple supervisory mechanism in other areas. The European Central Bank is the single supervisor of the monetary policy in the Euro Area. Considering these types of supervisory mechanisms, we have chosen the following variables for our study—GDP, $CO_2$ emissions, credit, inflation, non-performing loans, global financial crisis (dummy 1), and single supervisory mechanism (dummy 2).

Romania is outside the Euro Area and adopts the multiple supervisory mechanism, while being a member of the European Union. The Reserve Bank of India is the single supervisor of the national monetary policy, while the Brazilian National Monetary Council (CMN) is the single supervisor of banks, and of the monetary policy in Brazil. Regarding the latter, there are four regulators of the financial sector, such as—CVM for securities; the Central Bank for prudential and financial supervision; SUSEP for insurance; and PREVIC for pensions. These regulators share information between them and operate under the authority of the National Monetary Council, which includes the Ministry of Finance, Ministry of Planning, Budget and Management, and the Central Bank Governor. The CMN has no supervisory powers and it only issues guidelines. Moreover, in 2006, the Presidential Decree established a committee named COREMEC (Committee for the Regulation and Supervision of Financial, Securities, Insurance, and Complementary pension) under the Brazilian Ministry of Finance. At an international level, Brazil has been one of the signatories of the Basel Committee since 1988.

The countries were selected due to their differences in terms of GDP and population. Out of the three, in 2019, Romania had the highest GDP per capita ($12,943) and the lowest population (19.3 million). Regarding Brazil, its population and GDP per capita reached 209.4 million inhabitants and $8917. On the other hand, India had 1352.6 million citizens and a GDP per capita of $2010.

The research question of the present study revolved around the investigation of sustainable economic growth in the context of the global financial crisis and the single supervision mechanism.

More specifically, we analyzed various factors that impacted sustainable economic growth in the aforementioned frameworks.

This study provides an important contribution to the literature on banking supervisory mechanisms. Namely, the results showed that the global financial crisis did not have a direct impact on sustainable economic growth. To the best of our knowledge, these analyses mark the onset of studies on this topic, hence, our findings might stir the interest of other inquiries in this particular direction.

The paper comprises six sections. Section 1 includes an introduction into the research question, while the literature review belongs to Section 2. Section 3 draws on the research method, model, and data, and Section 4 reports the empirical results. Section 5 focuses on the discussion. Section 6 presents our concluding remarks, and the limitations of this paper with respect to future research paths and policy implications.

## 2. Literature Review

The supervisory mechanism assists sound bank management in order to facilitate financial stability. In turn, financial stability provides a stable growth for the national economy. The GDP growth rate indicates a sustainable economic growth at a country level but also at global level. The policies of financing implemented by banks indicate their lending behavior. In this context, macroprudential policies provide measures that aim at financial stability and the relationship between them is addressed in numerous studies. The interaction of macroprudential policies with financial stability and monetary policy does not assure a minimized impact of the global financial crisis. Therefore, price stability is not the answer to achieve macroeconomic stability, even though monetary policy and financial stability must focus on both price and output. Under these circumstances, a lack of coordination between monetary policy and fiscal policy enhanced the effects of the global financial crisis. An imperfect monetary policy provides incentives to correlate risks, particularly during expansions of the credit policy, in order to challenge recession. A sound monetary policy boosts the economy and increases the values of assets through lower rates. According to the IMF, an unsound monetary policy causes financial instability. Nier et al. (2011) suggested that effective coordination between the central bank and the supervisory authority is crucial for achieving financial stability. Since monetary policy governs financial stability (Ottmar 2003), central banks are assigned three main tasks, namely managing financial stability risks, acting as lenders of last resort, and supervising the payment and settlement system to achieve financial stability (Arrow et al. 2012).

In countries such as Argentina, Brazil, France, Italy, South Africa, or India, the central bank is the only supervisor that exclusively regulates banking activity.

After the global financial crisis caused the collapse of the financial system, central banks worked extensively to restore these systems all around the world. In light of this reality, the independence of a central bank is the key focus area for reforming efforts. Generally, macroprudential tools must be used to maintain financial stability, while the monetary policy should have price stability as its primary objective. One factor triggering the 2007–2009 global financial crisis was the faulty monetary policy.

In terms of the European Union and the Euro Area, there are three supervisory authorities such as the European Banking Authority, the European Insurance, and Occupational Pensions Authority and the European Securities Markets Authority. In addition, there is also the European Financial Stability Facility. Along the European Central Bank (ECB), each nation has its own national central bank. Across the European Union, a multiple supervisory mechanism is adopted, in order to establish financial and price stability. ECB plays the role of the custodian of the monetary policy in the European Union. Nevertheless, there are general concerns regarding the capability of the ECB to manage price stability and financial stability within the European Union and the Euro Area.

In order to achieve sustainable economic growth, green investment is essential. Therefore, the financing of green investment is supported with help from banking institutions. As a consequence, bank loans (i.e., credit) and the money supply are increasing, which might cause inflation, on the one hand, and a rise in the GDP, on the other hand. At the same time, the non-performance of loans

increases because of default repayments during a financial crisis. A higher GDP results in higher $CO_2$ emissions, which must be minimized. Green investments always target the long-term perspective, hence returns linked to such investments aim at the same time-frame. Moreover, the purpose of green investments is to transform a brown economy into a green one. In addition, green investments entail a highly advanced technology, which can be rather costly. Considering the financial resources demanded by a green economy, banks should act swiftly, so as to regulate the money supply and inflation, while taking into account sustainable economic growth. In order to meet the current challenges, the supervision of banks must be extremely efficient.

Sustainable economic growth reflects green growth, which is in line with the goals of sustainable development. Under this framework, $CO_2$ emissions represent the source for global warming, which causes adverse effects on world climate. Since the philosophy of green growth includes developmental dimensions such as social, economic, and sustainable use of natural resources, it is extremely important to mitigate $CO_2$ emissions within the limits of the Paris Treaty on Climate Change. Thus, the need for green investment generates more money supply within the economy. One of the financial investments in terms of loans and advances is represented by bank credits, which are directly associated with increased levels of default risks and rising inflation.

Air pollution increases with the expansion and development of urbanization (Liang et al. 2020). To achieve sustainable cities with less air pollution, policy makers must design such sustainable cities (Zhou et al. 2018). A case in point, India has introduced smart cities incorporating the model of sustainable cities and the Indian government performs annual review of such sustainable cities. Zhao et al. (2019) showed that the growth of PM 2.5 concentrations was proportionate to the growth of urbanization and industrialization. In the view of Singh et al. (2019), there is more economic growth in developed economies than developing economies, when there is higher renewable energy production. Moreover, the literature reports interesting studies according to which there is a relationship between aging and $CO_2$ emissions (Li et al. 2018). Namely, aging reduces productivity and, in turn, this indicates less consumption among the factors of production, such as the human capital and the natural capital.

Economic growth results in higher consumption of electricity, fossil fuels, and natural assets, which are challenges to sustainable economic growth (Zhao et al. 2016). A recent study reported that military expenditure in Romania had a negative impact on sustainable economic growth (Tao et al. 2020). Investigating economic growth in Romania with a focus on EU business climate, Hatmanu et al. (2020) revealed that, in the short run, the interest rate negatively influences economic growth, while the exchange rate influences it positively. In the EU, the monetary policy entails the effects of modifying the interest rate, which in turn influences price stability under the European System of Central Banks (ESCB) that includes the ECB and the national central banks of the European Union member states.

Dale et al. (2013) investigated carbon emissions based on the life-cycle data for electricity production and suggested environmental trade-offs in the use of large-scale renewables in Brazil. On this matter, the literature reports on the long-term relationship between financial development and economic growth (Shravani and Supran 2018). Guru and Yadav (2019) examined this relationship using banking sector and stock market indicators from emerging BRICS economies during the period of 1993–2014, and found a complementary link between the development of the banking sector and stock market (on one hand) and economic growth (on the other hand). Moreover, the impact of financial openness is larger than financial development in the post Asian crisis period for developing countries, than for the rest of the world (Estrada et al. 2010).

## 3. Research Method, Model and Data

The classical economic theory of Solow (1956) suggests that capital and labor contribute to economic growth. In the 1970s, social capital was considered a key contributor for achieving economic growth (Akcomak and Weel 2009; Coleman 1988; Helliwell 1996; Neira et al. 2009). Whitely (2002) examined 34 countries over the period 1970–1992 and his findings suggested that there was a strong relationship between social capital and economic growth. Benhabib and Spiegel (1994) also concluded

that human capital influenced economic growth. In terms of our research question, it is argued that green economy promotes economic growth, since it is inclusive and environmentally sustainable. Regarding this matter, the European Union, for instance, aims at adapting to the development policy of inclusive green economy, which ultimately generates growth, jobs creation, and poverty reduction, through a sustainable management of natural capital.

In our study, we conducted a panel data analysis using the statistical software EViews version 11, to study sustainable economic growth in India, Brazil, and Romania. Since Romania is a member of the European Union but does not belong to the Euro Area, our analyses included ten time-series data.

We considered the independent factors non-performing loans (NPL), bank credit, $CO_2$ emissions and inflation, and two dummy variables (i.e., dummy 1 = global financial crisis; dummy 2 = single supervisory mechanism), in order to investigate their influence on sustainable economic growth, even though there are several other factors that impact sustainable economic growth. For the purpose of this study, we focused on the abovementioned ones.

We used the Gross Domestic Product (GDP) as a proxy for sustainable economic growth and the period of analysis was 2005–2017. Data were drawn from the World Development Indicators (World Bank) and the Handbook of Statistics issued by the Reserve Bank of India.

## 4. Results

### 4.1. Descriptive Statistics

The descriptive statistics for the variables GDP, non-performing loans (NPL), $CO_2$ emissions, bank credit, inflation, global financial crisis (dummy 1), and single supervisory mechanism (dummy 2) are presented in Table 1.

**Table 1.** Descriptive statistics.

| Indicators | GDP | NPL | CO$_2$ Emissions | Credit | Inflation | Dummy 1 | Dummy 2 |
|---|---|---|---|---|---|---|---|
| Mean | 1.961176 | 6.481176 | 15.68000 | 144.5141 | 3.327059 | 0.423529 | 0.117647 |
| Median | 1.700000 | 4.300000 | 13.90000 | 140.6000 | 2.600000 | 0.000000 | 0.000000 |
| Maximum | 36.60000 | 33.80000 | 27.50000 | 1369.000 | 12.00000 | 1.000000 | 1.000000 |
| Minimum | −9.10000 | 0.700000 | 7.000000 | 20.60000 | −1.300000 | 0.000000 | 0.000000 |
| Std. Dev. | 5.309114 | 6.211845 | 5.392119 | 144.3489 | 2.826043 | 0.497050 | 0.324102 |
| Skewness | 3.071751 | 2.446125 | 0.922380 | 7.266090 | 1.032640 | 0.309524 | 2.373464 |
| Kurtosis | 22.73131 | 9.581283 | 2.862901 | 62.43425 | 3.700951 | 1.095805 | 6.633333 |
| Jarque-Bera | 1512.529 | 238.1678 | 12.11936 | 13258.64 | 16.84669 | 14.19917 | 126.5595 |
| Probability | 0.000000 | 0.000000 | 0.002335 | 0.000000 | 0.000220 | 0.000825 | 0.000000 |
| Sum | 166.7000 | 550.9000 | 1332.800 | 12283.70 | 282.8000 | 36.00000 | 10.00000 |
| Sum Sq. Dev. | 2367.682 | 3241.310 | 2442.296 | 1750274. | 670.8678 | 20.75294 | 8.823529 |

Therefore, one can see the average values of the variables considered in the study—196.11% for GDP; 648.11% for NPL; 1568% for $CO_2$ emissions; 14,451.41% for credit; and 332.70% for inflation. On the other hand, the median values for each indicator were—170% for GDP; 430% for NPL; 1390% for $CO_2$ emissions; 1406% for credit; and 260% for inflation. It was observed that the median values were close to the mean values of the variables included in our analyses. This meant that 50% of the data took values below the median and 50% took values above the median.

The value of the standard deviation suggests a more accurate and detailed estimate of the dispersion. Moreover, standard deviations indicate the fluctuation of the time-series. In this sense, the variable credit had the largest volatility, followed by NPL, while dummy 2 had the smallest volatility.

The positive skewness values showed that all variables were skewed to the right. More specifically, the skewness of variables NPL, $CO_2$ emissions, inflation, dummy 1, and dummy 2 were less than three.

The kurtosis values for GDP, NPL, credit, inflation, and dummy 2 were above three, which indicated leptokurtic distributions. Hence, the dataset had a lighter tail than the normal distribution. Since the kurtosis of the $CO_2$ emissions variable was below three, it meant that its distribution was platykurtic. The high significant values of the Jarque-Bera test indicates that our variables of interest were non-normally distributed at the 1% level.

### 4.2. Pooled OLS, Fixed Effect, and Random Effect Model

As mentioned by Baltagi (2008), any empirical analysis should start with the decision of estimating results with a panel regression or a simple regression. For this purpose, one should run a specific test that assists such a decision. Our first results obtained in EViews suggested that the null hypothesis, according to which the individual effects were null, had to be rejected, since the OLS estimator was unfit and inconsistent. Table 2 shows estimates of the pooled regression, fixed effect model, and random effect model.

$$GDP_{it} = \beta_1 + \beta_2 NPL + \beta_3 Credit + \beta_4 CO2 + \beta_5 Inflation + \beta_6 CO2 + \epsilon_{it} \quad (1)$$

**Table 2.** Pooled regression model, fixed effect model, and random effect model.

| Variable | Coefficient | Std. Error | *t*-Statistic | Prob. | Obs. |
|---|---|---|---|---|---|
| **Pooled Regression** | | | | | |
| C | −6.242789 | 2.437462 | −2.561185 | 0.0123 | 85 |
| NPL | 0.185826 | 0.096485 | 1.925946 | 0.0577 | 85 |
| Credit | 0.001226 | 0.004013 | 0.305583 | 0.7607 | 85 |
| $CO_2$ emissions | 0.436957 | 0.149829 | 2.916371 | 0.0046 | 85 |
| Inflation | −0.008744 | 0.284945 | −0.030685 | 0.9756 | 85 |
| **Fixed Effect** | | | | | |
| C | −15.21163 | 7.416282 | −2.051113 | 0.0439 | 85 |
| NPL | 0.264866 | 0.147383 | 1.797125 | 0.0766 | 85 |
| Credit | −0.000909 | 0.004259 | −0.213520 | 0.8315 | 85 |
| $CO_2$ emissions | 0.990180 | 0.455954 | 2.171667 | 0.0332 | 85 |
| Inflation | 0.018504 | 0.454362 | 0.040726 | 0.9676 | 85 |
| **Random Effect** | | | | | |
| C | −6.242789 | 2.444442 | −2.553870 | 0.0126 | 85 |
| NPL | 0.185826 | 0.096762 | 1.920446 | 0.0584 | 85 |
| Credit | 0.001226 | 0.004024 | 0.304710 | 0.7614 | 85 |
| $CO_2$ emissions | 0.436957 | 0.150258 | 2.908042 | 0.0047 | 85 |
| Inflation | −0.008744 | 0.285761 | −0.030597 | 0.9757 | 85 |

According to Table 2, the factor $CO_2$ emissions positively influenced the dependent variable of sustainable economic growth by 43.6% in the pooled regression and random effect models, while in the fixed effects model it significantly increased sustainable economic growth by 99%.

The next step in choosing between the fixed effect model and the random effect model consisted of running the Hausman test. In this case, the null hypothesis would imply that there were no significant differences between the estimates of the fixed effect model and the random effect model. If the null hypothesis was rejected, the fixed effect model should be chosen. Otherwise, the random effect model would be considered to be more adequate. Table 3 shows the output of this test.

**Table 3.** The Hausman test.

| Test Summary | Chi-Sq. Statistic | | Chi-Sq. d.f. | | Prob. | |
|---|---|---|---|---|---|---|
| Cross Section Random | 6.653947 | | 4 | | 0.1553 | |
| | **Fixed Effect** | **Random Effect** | **Var. (Diff.)** | **Prob.** | **Obs.** | |
| C | | | | | 85 | |
| NPL | 0.264866 | 0.185826 | 0.012359 | 0.4771 | 85 | |
| Credit | −0.000909 | 0.001226 | 0.000002 | 0.1260 | 85 | |
| $CO_2$ | 0.990180 | 0.436957 | 0.185316 | 0.1988 | 85 | |
| Inflation | 0.018504 | −0.008744 | 0.124786 | 0.9385 | 85 | |

As one can see from Table 3, the *p*-value was above 0.05, therefore, the null hypothesis was not rejected and we could conclude that the random effect model was more suitable for our study. According to the random effect model, the variable $CO_2$ emissions contributed annually to the increase in the economic growth, proxied by GDP, with 43.6%. In other words, when there are more manufacturing and agricultural activities supporting a higher economic growth, the increasing $CO_2$ emissions need to be regulated in order to achieve a sustainable economic growth.

### 4.3. Unit Root and Hadri Test

In the case of a panel, the unit root test was conducted to investigate each individual series for stationarity. The null hypothesis assumed nonstationary series, while the alternate hypothesis assumed a stationary series. In other words, the mean, variance, and autocorrelation structure remained unchanged over the entire time-frame. When a time-series was stationary, this aspect could be changed to nonstationary, through techniques like the first or second difference.

$$\Delta Y_t = \beta y_{t-1} + \varepsilon \tag{2}$$

$$\Delta Y_t = b_0 + \beta y_{t-1} + \varepsilon \tag{3}$$

$$\Delta Y_t = b_0 + \beta y_{t-1} + b_2 + \varepsilon \tag{4}$$

$$\text{Null Hypothesis} = H_0 \ : \beta = 0 \tag{5}$$

$$\text{Alternate Hypothesis} = H_a \ : \ \beta \ < \ 0 \tag{6}$$

Therefore, we conducted a panel unit root test (Table 4) to investigate the link between sustainable economic growth and the independent variables NPL, credit, $CO_2$ emissions, inflation, global financial crisis, and the single supervisory mechanism. In this context, the Levin, Lin, and Chu test was useful to conduct a unit root test when different individual aspects were integrated into a final regression. According to the results in Table 4, except for the independent variable NPL, the null hypothesis of the unit root was rejected for all the other variables. Therefore, we could conclude that the series had no unit root and it was stationary.

**Table 4.** Panel Unit Root test.

| Variables | Panel Unit Root | | | | |
|:---:|:---:|:---:|:---:|:---:|:---:|
| | Method: Levin, Lin, and Chu Test | | | | |
| | Statistic | Prob. | Cross-Sections | Obs. | Hypothesis |
| GDP | −7.34874 | 0.0000 | 10 | 101 | Null: rejected |
| D (GDP) | −11.5736 | 0.0000 | 10 | 101 | Null: rejected |
| NPL | −1.61694 | 0.0529 | 10 | 104 | Null: not rejected |
| D(NPL) | −2.66854 | 0.0038 | 10 | 92 | Null: rejected |
| Credit | −5.56547 | 0.0000 | 10 | 116 | Null: rejected |
| D(Credit) | −4.20165 | 0.0000 | 10 | 105 | Null: rejected |
| $CO_2$ | −3.33351 | 0.0004 | 10 | 88 | Null: rejected |
| $D(CO_2)$ | −8.83459 | 0.0000 | 10 | 76 | Null: rejected |
| Inflation | −4.00495 | 0.0000 | 10 | 119 | Null: rejected |
| D (Inflation) | −9.13087 | 0.0000 | 10 | 108 | Null: rejected |

Table 5 displays the results of the Hadri Unit Root test. According to the test, the null hypothesis was not rejected as $p > 0.05$. Again, we can conclude that there was stationarity in the series.

**Table 5.** Hadri Unit Root test.

| Method | Statistic | Prob. |
|:---:|:---:|:---:|
| Hadri Z-statistic | −0.32742 | 0.6283 |
| Heteroscedastic consistent Z-statistic | 2.47301 | 0.0067 |

Note: Series GDP; Sample: 2005–2017; Observations: 129; Null Hypothesis: Stationarity.

*4.4. Vector Error Correction Model (VECM)*

To estimate this model, data should be considered on a number of time-periods, $t = 1, \cdots, T$, and a number of groups, $I = 1, \cdots, N$.

The model was as follows:

$$y_t = A_1 y_{t-1} + A_2 y_{t-2} + \ldots + A_p y_{t-P} + B_{x_t} + \mu_t \tag{7}$$

$$y_t = A_1 y_{t-1} + A_2 y_{t-2} + \ldots + A_p y_{t-P} + \mu_t \tag{8}$$

$$\Delta y_t = \Pi\, y_{t-1} + \sum_{i=1}^{P-1} T_i \Delta y_{t-i} + \mu_t \tag{9}$$

where

$$\Pi = \sum_{i=1}^{P} A_i - I\Gamma_i = \sum_{j=i+1}^{P} A_j$$

The formula of cointegration relationship was written as:

$$\Delta y_t = \alpha \beta' y_{t-1} + \sum_{i=1}^{P-1} T_i \Delta y_{t-i} + \mu_t$$

Therefore, the VECM model was written as follows:

$$\Delta y_t = \alpha ecm_{t-1} + \sum_{i=1}^{P-1} T_i \Delta y_{t-i} + \mu_t$$

### 4.4.1. Panel Least Squares Method

Regarding the Panel Least Squares method (Table 6), results indicated that three variables significantly influenced sustainable economic growth, namely non-performing loans (NLP), $CO_2$ emissions, and single supervisory mechanism (dummy 2). Hence, when NLP and $CO_2$ emissions increased by 10%, GDP rose by 10.11% and 44.3%, respectively. The *p*-value was 2%, which was below the 5% level. In other words, there was a long-term equilibrium between the variables considered. Moreover, this model was statistically significant.

**Table 6.** Panel Least Square Method.

| | Dependent Variable: GDP Number of Observations: 85 | | | |
|---|---|---|---|---|
| | **Coefficient** | **Std. Error** | ***t*-Statistic** | **Prob.** |
| C(1) | −17.58493 | 7.394438 | −2.378129 | 0.0202 |
| C(2) | 0.260220 | 0.146946 | 1.770848 | 0.0810 |
| C(3) | −0.000572 | 0.004159 | −0.137460 | 0.8911 |
| C(4) | 1.011893 | 0.443858 | 2.279770 | 0.0257 |
| C(5) | 0.509788 | 0.486720 | 1.047395 | 0.2986 |
| C(6) | −0.334251 | 1.189069 | −0.281103 | 0.7795 |
| C(7) | 4.430069 | 1.941081 | 2.282270 | 0.0256 |

### 4.4.2. The Wald Test

Furthermore, we conducted the Wald test, in order to choose between the pooled effect model and the fixed effect model. As the null hypothesis was accepted, we concluded that the pooled regression was adequate in our case.

The results of the Wald test (Table 7) showed that our variables also established a short-term equilibrium between them and that the model was statistically significant.

**Table 7.** The Wald test.

| **Test Statistic** | **Value** | **d.f.** | **Prob.** |
|---|---|---|---|
| *t*-statistic | −2.411005 | 69 | 0.0186 |
| *F*-statistic | 5.812948 | (1.69) | 0.0186 |
| *Chi*-square | 5.812948 | 1 | 0.0159 |

### 4.4.3. Johansen Fisher Panel Cointegration Test

$$y_t = \delta + \theta y_{t-1} + \varepsilon_t \tag{10}$$

$$\Delta y_t = \pi(y_{t-1} - \mu) + \sum_{i=1}^{p-1} y_i \Delta y_{t-1} + \varepsilon_t \tag{11}$$

where

$$\Pi = \sum_{i=1}^{P} \theta_i - IT_i = -\sum_{j=i+1}^{P} \theta_j$$

Null Hypothesis = $H_0 : \beta = 0$

Alternate Hypothesis = $H_a : \beta < 0$

Table 8 shows the results of the Johansen Fisher panel cointegration test, which estimated the restricted or unrestricted VECM, in order to identify the short-term and long-term relationship between variables. This procedure is important for the estimation of the error correction models.

**Table 8.** Johansen Fisher panel cointegration test.

| Hypothesized No. of CE(s) | Fisher Stat. (from Trace Test) | Prob. | Fisher Stat. (from Max-Eigen Test) | Prob. | Obs. |
|---|---|---|---|---|---|
| Series GDP–$CO_2$ | | | | | |
| None | 129.3 | 0.0000 | 105.3 | 0.0000 | 130 |
| At most 1 | 67.68 | 0.0000 | 67.68 | 0.0000 | 130 |
| Series GDP Inflation | | | | | |
| None | 38.65 | 0.0074 | 31.73 | 0.0462 | 130 |
| At most 1 | 37.52 | 0.0101 | 37.52 | 0.0101 | 130 |
| Series: GDP Credit | | | | | |
| None | 85.43 | 0.0000 | 60.89 | 0.0000 | 130 |
| At most 1 | 68.50 | 0.0000 | 68.50 | 0.0000 | 130 |
| Series: GDP NPL | | | | | |
| None | 163.2 | 0.0000 | 164.6 | 0.0000 | 130 |
| At most 1 | 38.47 | 0.0077 | 38.47 | 0.0077 | 130 |

When GDP and the $CO_2$ emissions series were considered in the Johansen Fisher panel cointegration test, we concluded that there was cointegration, since the null hypothesis was rejected both in the case of "none" and "at most 1". The *p*-value for the Fisher statistic (from trace test) was below 0.001 and it also remained the same for the value from the max-eigen test. In the case of the GDP and inflation series, the null hypothesis was rejected in both "none" and "at most 1". Hence, one could state that there was cointegration in more than one series, which was emphasized by the *p*-value of both Fisher statistics. The same conclusions apply for the GDP and credit series, but also for the GDP and the NPL series. Consequently, the VECM model was significant.

As shown by Table 9, in the case of the Indian data, there is at least one cointegration relationship associated with the series GDP–credit and at most one relationship corresponding to the series GDP–NPL, GDP–inflation, and GDP–$CO_2$ emissions.

**Table 9.** Individual cross-section results—India.

| Hypothesized Cointegration | Trace Test Statistics | Prob. | Max-Eigen Test Statistics | Prob. | Obs. |
|---|---|---|---|---|---|
| GDP NPL | | | | | |
| None | 22.4503 | 0.0038 | 20.3207 | 0.0049 | 130 |
| At most 1 | 2.1296 | 0.1445 | 2.1296 | 0.1445 | 130 |
| Series GDP Inflation | | | | | |
| None | 12.3876 | 0.1393 | 11.3700 | 0.1366 | 130 |
| At most 1 | 1.0176 | 0.3131 | 1.0176 | 0.3131 | 130 |
| Series: GDP Credit | | | | | |
| None | 23.3224 | 0.0027 | 18.9681 | 0.0084 | 130 |
| At most 1 | 4.3542 | 0.0369 | 4.3542 | 0.0369 | 130 |
| Series: GDP $CO_2$ emissions | | | | | |
| None | 15.5125 | 0.0497 | 12.2569 | 0.1014 | 130 |
| At most 1 | 3.2556 | 0.0712 | 3.2556 | 0.0712 | 130 |

Table 10 displays the results from the Brazilian data. As can be observed, the series GDP–inflation has a minimum of one cointegration relationship, while the series GDP–NPL, GDP–credit, and GDP–$CO_2$ emissions had at most one relationship (*p*-values were above 0.05).

**Table 10.** Individual cross-section results—Brazil.

| Hypothesized Cointegration | Trace Test Statistics | Prob. | Max-Eigen Test Statistics | Prob. | Obs. |
|---|---|---|---|---|---|
| GDP NPL | | | | | |
| None | 10.5782 | 0.2388 | 7.5466 | 0.4267 | 130 |
| At most 1 | 3.0316 | 0.0817 | 3.0316 | 0.0817 | 130 |
| Series GDP Inflation | | | | | |
| None | 14.9958 | 0.0593 | 9.9906 | 0.2126 | 130 |
| At most 1 | 5.0052 | 0.0253 | 5.0052 | 0.0253 | 130 |
| Series: GDP Credit | | | | | |
| None | 7.4983 | 0.5205 | 6.4030 | 0.5621 | 130 |
| At most 1 | 1.0953 | 0.2953 | 1.0953 | 0.2953 | 130 |
| Series: GDP $CO_2$ emissions | | | | | |
| None | 37.2419 | 0.0000 | 36.8337 | 0.0000 | 130 |
| At most 1 | 0.4082 | 0.5229 | 0.4082 | 0.5229 | 130 |

As shown by Table 11, in the case of Romania, there was at least one cointegration relationship regarding the series GDP–credit, GDP–$CO_2$ emissions, and GDP–NPL as the *p*-values were below 0.05 and the null hypothesis was rejected. We can conclude that sustainable economic growth was significantly influenced by bank credit, the level of $CO_2$ emissions, and the non-performing loans.

**Table 11.** Individual cross-section results—Romania.

| Hypothesized Cointegration | Trace Test Statistics | Prob. | Max-Eigen Test Statistics | Prob. | Obs. |
|---|---|---|---|---|---|
| GDP NPL | | | | | |
| None | 12.2504 | 0.1453 | 8.3071 | 0.3483 | 130 |
| At most 1 | 3.9433 | 0.0471 | 3.9433 | 0.0471 | 130 |
| Series GDP Inflation | | | | | |
| None | 15.0782 | 0.0577 | 13.9839 | 0.0553 | 130 |
| At most 1 | 1.0943 | 0.2955 | 1.0943 | 0.2955 | 130 |
| Series: GDP Credit | | | | | |
| None | 28.4346 | 0.0003 | 23.2053 | 0.0015 | 130 |
| At most 1 | 5.2292 | 0.0222 | 5.2292 | 0.0222 | 130 |
| Series: GDP $CO_2$ emissions | | | | | |
| None | 35.6170 | 0.0000 | 23.8899 | 0.0011 | 130 |
| At most 1 | 11.7272 | 0.0006 | 11.7272 | 0.0006 | 130 |

*4.5. Panel Fully Modified Least Squares (FMOLS)*

$$y_t = Ax_t + u_a$$

In addition, we conducted a Panel Fully Modified Ordinary Least Squares (FMOLS) test, a non-parametric technique that deals with serial correlation (Table 12).

**Table 12.** Panel Fully Modified Ordinary Least Squares (FMOLS).

| Dependent Variable: GDP Long-Run Covariance Estimates (Bertett Kernel, Newey-West Fixed Bandwidth) | | | | |
|---|---|---|---|---|
| **Variable** | Coefficient | Std. Error | t-Statistic | Prob. |
| Credit | 0.097374 | 0.109659 | 0.887965 | 0.3780 |
| NPL | −0.894416 | 0.461323 | −1.938806 | 0.0572 |
| $CO_2$ emissions | 4.486095 | 0.611393 | 7.337502 | 0.0000 |
| Inflation | 1.295877 | 0.556299 | −2.329463 | 0.0232 |
| R-squared | −293.682737 | | | |
| **Long-Run Variance** | 5.698580 | | | |
| Credit | −0.074803 | 0.015279 | −4.895911 | 0.0000 |
| NPL | −1.090063 | 0.275363 | −3.958644 | 0.0002 |
| $CO_2$ emissions | 0.371321 | 0.099364 | 3.736976 | 0.0004 |
| Inflation | 0.789796 | 0.162911 | 4.848035 | 0.0000 |
| Dummy 1 | −0.553117 | 0.259445 | −2.131926 | 0.0366 |
| Dummy 2 | 5.293318 | 0.619588 | 8.543284 | 0.0000 |
| R-squared | −10.876035 | | | |
| **Long-Run Variance** | 0.7966630 | | | |

As can be seen from Table 12, we estimated two FMOLS models. The first model included only the independent variables credit, NPL, $CO_2$ emissions, and inflation, while the second model included the independent variables, together with the dummy variables of global financial crisis (dummy 1) and

single supervisory mechanism (dummy 2). The R-squared for the first model was −293.68, while the R-squared for the second model was −10.87.

Regarding the first FMOLS model, results confirmed that the variables $CO_2$ emissions and inflation significantly influenced sustainable economic growth by 448% and 129%, respectively, since the *p*-values for both variables were below 0.05.

In terms of the second FMOLS model also comprising the two dummy variables, it can be observed that all variables considered had a significant impact on sustainable economic growth because all *p*-values were less than 0.05. Nevertheless, the variables registered mixed speeds of adjustment. Hence, the factors credit, non-performing loans, and the global financial crisis triggered a negative speed of adjustment, while the level of $CO_2$ emissions, inflation, and the single supervisory mechanism (SSM) triggered a positive speed of adjustment.

In addition, we conducted the FMOLS with a focus on the impact of dummy variables, as can be seen from Table 13.

**Table 13.** Panel Fully Modified Ordinary Least Squares.

| Dependent Variable: GDP; Series: GDP, NPL, Credit, $CO_2$ Emissions, Inflation, Dummy 1 and Dummy 2 | | | | |
|---|---|---|---|---|
| **Variable** | **Coefficient** | **Std. Error** | ***t*-Statistic** | **Prob.** |
| C | −5.477981 | 2.594342 | −2.111510 | 0.0379 |
| NPL | 0.142985 | 0.096896 | 1.475652 | 0.1441 |
| Credit | 0.001641 | 0.003941 | 0.416357 | 0.6783 |
| $CO_2$ | 0.331795 | 0.153464 | 2.162043 | 0.0337 |
| Inflation | 0.232887 | 0.295033 | 0.789361 | 0.4323 |
| Dummy 1 | −0.437483 | 1.151825 | −0.379817 | 0.7051 |
| Dummy 2 | 4.107253 | 1.820715 | 2.255846 | 0.0269 |
| R-squared | 0.2232 | | | |
| Prob. (*F*-statistic) | 0.002552 | | | |

Since the *p*-value of the FMOLS model was below 0.01, we could conclude that it was relevant. As indicated by the corresponding *p*-values (3.37% and 2.69%, respectively), both the level of $CO_2$ emissions and the single supervisory mechanism had a significant influence on sustainable economic growth, proxied by GDP. Therefore, it could be concluded that the sustainable economic growth increased by 33% when the level of $CO_2$ emissions was augmented by 100%. Moreover, it also increased by 410%, following a change in the single supervisory mechanism. The impact of the global financial crisis was not significant.

## 5. Discussion

This research study was conducted with the purpose of investigating the variables impacting on sustainable economic growth, with a special focus on the global financial crisis and the single supervisory mechanism. To the best of our knowledge, this is the first study tackling such a direction. The country sample included Brazil, India, and Romania, each of which is a member of an important regional (i.e., European Union) or global organization (i.e., BRICS). We chose the 2005–2017 time-frame for our analyses, in order to compare the state of national economies before, during, and after the global financial crisis. As a methodological approach, we favored panel data analysis conducted with the EViews statistical package version 11.

Our results were in line with the existing literature on sustainable economic growth. A case in point, Bargaoui and Nouri (2017) carried out a dynamic panel data analysis regarding $CO_2$ emissions of 114 nations, during the period 1980–2010, and their results indicated that $CO_2$ emissions level was directly related to changes in sustainable economic growth. Aye and Edoja (2017) analyzed the

impact of economic growth on $CO_2$ emissions in 31 developing countries, using a dynamic panel threshold model for data regarding the period of 1970–2013. Their results showed that economic growth negatively influenced $CO_2$ emissions in low growth systems, but positively impacted it in high growth systems. Their findings were critical in the light of recent realities, according to which countries such as India and China, which are experiencing intensive economic growth, were responsible for the production of massive carbon dioxide emission levels.

According to our results, carbon dioxide emissions continued to prevail annually in every 33% of the economic growth level. This constituted an important finding that should be taken into account when designing national and regional economic policies, which should be in line with market realities. We deem that our results might be valid for both developed and developing nations.

Moreover, an effective banking supervisory mechanism considerably influenced the level of sustainable economic growth (i.e., by 410%). This second result should also be considered in order to enact effective supervisory mechanisms. The literature reports similar findings in this direction. For instance, by using data from 23 OECD countries, Guru and Yadav (2019) found that policy makers should formulate effective financial sector policies to promote sustainable economic growth.

After running a pooled regression model, a fixed effect model, and a random effect model, we concluded that $CO_2$ emissions affected sustainable economic growth. On this matter, Leitão (2010) examined the countries included in the EU-27 and the BRIC association (i.e., Brazil, Russia, India, China) and reported that the general opinion according to which financial development stimulates economic growth did not apply during financial crises. His findings were in line with our results. Moreover, Popov (2017) collected evidence to establish that credit availability was crucial for economic growth.

During and after the 2007–2009 global financial crisis, factors such as bank credit, inflation, and non-performing loans negatively impacted sustainable economic growth. Consequently, our study offered important insights into the evolution of sustainable economic growth in emerging economies. Our endeavor was important because during the global financial crisis, traditional economic doctrines apparently failed to provide adequate solutions for mitigating the economic turmoil. For instance, due to the fact that the Euro Area had a single supervisory mechanism, several countries were affected by the crisis (e.g., Greece, Italy). Nevertheless, EU countries recently began to invest heavily in green finance, in order to boost demand and supply.

On the other hand, India continued to expand its economy with a higher growth rate. Hence, there are various aspects that need to be considered and managed through effective supervision mechanism of banks, in order to prevent future financial and economic downturns. In this sense, Shastri et al. (2018) examined fiscal sustainability in India, Pakistan, Bangladesh, Sri Lanka, and Nepal, during the period of 1985–2014. Their results indicated that a weak fiscal sustainability should be tackled through a long-term fiscal discipline. Therefore, the banking supervisory mechanism should design suitable methods and policies to achieve such fiscal discipline.

According to our results, the variables GDP, $CO_2$ emissions, non-performing loans, bank credit, and inflation have more than one cointegration relationship among them. Our variables collectively influenced sustainable economic growth. However, when considering individual cross-section results, differences arise from one nation to another, as expected. In the case of Romania, GDP, credit, and $CO_2$ emissions register more than one cointegration relationship. Regarding the Brazilian data, the variables GDP, inflation, and $CO_2$ emissions share more than one cointegration relationship. As for the data in India, GDP, non-performing loans, $CO_2$ emissions, bank credit, and inflation have one cointegration relationship. The variable inflation did not influence sustainable economic growth in most countries, as also reported by Xiao (2009). Moreover, Aydin (2017) did not find direct evidence that inflation impacted on economic growth, although price stability was the preferred economic state.

## 6. Conclusions and Policy Implications

The scientific literature reports numerous studies on economic growth. For instance, according to Leitão (2010), factors like banks, credit, consumer price, imports and exports, and productivity influence

economic growth. Analyses ran on data from 23 OECD countries (including several EU members) showed that financial development plays an important role in economic growth (OECD 2014).

To the best of our knowledge, there is no evidence to indicate that the global financial crisis negatively impacted on sustainable economic growth. A case in point, India continued to achieve high economic growth during the entire period of study. Therefore, the countries from our sample pool should apply the subsequent policies in order to boost sustainable economic growth. India needs to revive bank credit within the corporate sector and to rationalize non-performing loans, in order to solve the current challenges of its economy. Namely, the demonetization of the national currency affected its economic growth and the country was asked to improve its fiscal and monetary policies to boost sustainable economic growth. As for Romania, the country needs to expand its networks of national banks and foreign banks for better financial and non-financial services, with the aim of achieving a sustainable economic growth. For the same purpose, Brazil should introduce an effective single supervisory mechanism.

Sustainable economic growth is a desirable goal for every economy, as it helps to implement the Paris Agreement on global warming. For this reason, the levels of $CO_2$ emissions pose a great challenge to all nations around the world in general and to our country sample, in particular. As indicated in our results, the $CO_2$ emissions significantly influenced the sustainable economic growth, and so did the single supervisory mechanism. As a consequence, authorities from these countries should design adequate economic policies for the long run.

Technology helps reducing energy consumption, which in turn mitigates $CO_2$ emissions (Yin et al. 2020). Romania, for instance, should encourage green investments in order to reduce pollution levels. The country established a favorable business environment to develop corporate social responsibility, human capital, and financial and non-financial performance (Barrena-Martinez et al. 2019; Gangi et al. 2019).

Our results differ from the findings of Aydin and Obadasioglu (2017), who used a threshold autoregressive model to study the relationship between financial development and economic growth. Their study indicated that low and moderate inflation influenced economic growth in Romania and Turkey. Moreover, findings from Aye and Edoja (2017) indicated that $CO_2$ emissions varied from a low growth system to a high growth system, which differed from our findings.

The current study had some limitations. The first limitation was the non-availability of data regarding $CO_2$ emissions and non-performing loans for the period 2015–2017, which however, did not significantly influence our results. The second limitation was that no detailed analyses on the implications of $CO_2$ emissions and single supervisory mechanisms were available for individual countries. The third limitation was that GDP (chosen as proxy) and sustainable economic growth varied in certain aspects but also shared common grounds. We opted to use GDP as a proxy for this phenomenon as it captured the magnitude of economic growth for all countries across the entire time-frame.

The current study opens an engaging conversation into the literature investigating economic growth and the factors influencing it. Our results offer important insights about the impact of the global financial crisis and the banking supervisory mechanism on sustainable economic growth. For this reason, we deem that our study could represent the beginning for other scientific investigations in this regard and might lay the foundation for future research.

**Author Contributions:** Conceptualization, B.I. (Batrancea Ioan); methodology, B.I. (Batrancea Ioan); software, G.L.; validation, R.M.K.; formal analysis, R.M.K.; resources, F.G.; data curation, B.I. (Bircea Ioan); writing—original draft preparation, R.M.K.; writing—review and editing, B.L. and N.A.; visualization, T.H.; supervision, R.M.-I.; and project administration, B.I. (Batrancea Ioan). All authors have read and agreed to the published version of the manuscript.

**Funding:** This research was funded by the Babes-Bolyai University of Cluj-Napoca through the Grants for Supporting Employees' Competitiveness AGC–32865/26.07.2019, AGC–33374/12.09.2019 and AGC–30121/17.01.2020.

**Acknowledgments:** The authors gratefully thank the editors and the anonymous reviewers of the journal for their useful and constructive comments, which improved the quality of the paper.

**Conflicts of Interest:** The authors declare no conflict of interest.

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
