# Peer review of "A Panel Data Analysis on Sustainable Economic Growth in India, Brazil, and Romania"

_jrfm, doi:10.3390/jrfm13080170_

Round 1

Reviewer 1 Report

  1. The authors do not set forth in the paper the essence and content of the fundamental concept of economic science “sustainable growth of a country's economy” or “sustainable economy”. This leads to errors in the selection of indicators.
    The sustainable economics encompasses four basic system principles: (a) maintaining the health of ecosystems and the livelihoods that they provide to people; (b) the extraction of renewable resources (such as fish or wood) at a speed not exceeding the rate of their recovery; (c) the consumption of non-renewable resources (such as fossil fuels and minerals) at a speed that allows them to be replaced by renewable counterparts before the non-renewable resource is exhausted; (d) deposit of waste in the environment at a rate not exceeding the rate of safe assimilation.
    An example of this kind of research is the model of Canadian economist Peter A. Victor.( Peter A. Victor http://www.pvictor.com)
  2. The authors are mistaken that the steady growth of the country's economy can be assessed by one indicator. This statement follows from the definition of the concept of “ The Sustainable Economy”.
  3. Panel analysis of data using EViews statistical software has been excellent.

Author Response

Reply to Reviewer 1

Dear Colleague,

We are extremely grateful for your availability in reviewing this article, which has been resubmitted in revised version for publication. Your comments were very helpful in substantially improving the quality of the paper. Please find below the answers to your comments in a point-by-point manner.

  1. The authors do not set forth in the paper the essence and content of the fundamental concept of economic science “sustainable growth of a country's economy” or “sustainable economy”. This leads to errors in the selection of indicators.
    The sustainable economics encompasses four basic system principles: (a) maintaining the health of ecosystems and the livelihoods that they provide to people; (b) the extraction of renewable resources (such as fish or wood) at a speed not exceeding the rate of their recovery; (c) the consumption of non-renewable resources (such as fossil fuels and minerals) at a speed that allows them to be replaced by renewable counterparts before the non-renewable resource is exhausted; (d) deposit of waste in the environment at a rate not exceeding the rate of safe assimilation.
    An example of this kind of research is the model of Canadian economist Peter A. Victor.( Peter A. Victor
    http://www.pvictor.com)

Reply: Thank you for this suggestion. We have suitably clarified and incorporated the essence and content of the sustainable economic growth concept in the revised article. Moreover, Victor’s concept and model are discussed in the paper. Please find this change on page 3 of the revised version.

  1. The authors are mistaken that the steady growth of the country's economy can be assessed by one indicator. This statement follows from the definition of the concept of “The Sustainable Economy”.

Reply: Thank you for your observation. We have included CO2 as one of the factors assessing the economic behavior of a nation that aims sustainable economic growth. The other independent factors considered (i.e., credit, inflation, non-performing loans) are equally important in this regard.

  1. Panel analysis of data using EViews statistical software has been excellent.

Reply: Thank you very much for your appreciation. We are extremely grateful for this comment.  

Reviewer 2 Report

RƏY

A Panel Data Analysis on the Sustainability of Economic Growth in India, Brazil and Romania

  1. It is good if you write the abstract again.You can view abstracts written in articles published in any MDPI journal.
  2. Keywords should be changed, for example: CO2 emissions, panel data analysis, GDP, Sustainability of Economic Growth, global financial crisis, credit, inflation, banking supervisory mechanisms.
  3. “Banks play an important role in providing credit for green investment to mitigate CO2 emissions and achieve the sustainability of economic growth. As a result of increased credit, money supply multiplies, which in turn adds to inflation and any default in repayment results in non-performance of assets (known as non-performance of loans or NPL). On the other hand, higher investment leads to economic growth and it results in higher growth rate of the gross domestic product (GDP).” (76-81) and “Financing green investment is achieved with help from banking institutions. Therefore, bank loans (i.e., credit) are increasing and the money supply multiplies, which leads to inflation on the one hand and the growth of the GDP on the other hand.” (160-162) In my opinion, this should not be peculiar to loans only for green investments. It does not cause inflation, it acts as one of reasons for inflation. After all, the volume of green investments is not so much.
  4. “Brazil had achieved higher economy growth rates during the period 2003-2010, but this drastically has declined from 2011 onwards due to the impact of the global financial crisis”.(57-59) The world financial crisis of 2008-2009 more precise.
  5. “To the best of our knowledge, this appears to be the first research paper on this topic and we firmly believe that our findings will encourage further research in this direction”. (115-116)“…we firmly believe that our” change.
  6. “According to specialists, the ineffective coordination of the two deepened the most recent global financial crisis”. (126-127) Who are these specialists?
  7. “Green economy promotes economic growth, which is inclusive and environmentally sustainable” (155-156) “In order to achieve the sustainability of economic growth, green investment is essenal” (159-160) Make adjustments.
  8. The first limitation is the non-availability of data regarding CO2 emissions and non-performing loans for the period 2015–2017. (435-437) In this case, How was the study conducted?
  9. From the literature I mentioned below, in the “Literature review”, “Discussion”, “Conclusion and policy implications”. Use.
  10. The article is good. The mentioned comments are quickly eliminated by specialists like you. I think the opinions of my and other reviewers are taken into account. Good luck to you.

Recommended literature:

  1. Liang, Z.; Wei, F.; Wang, Y.; Huang, J.; Jiang, H.; Sun, F.; Li, S. The Context-Dependent Effect of Urban Form on Air Pollution: A Panel Data Analysis. Remote Sens.202012, 1793.
  2. Zhao, H.; Guo, S.; Zhao, H. Quantifying the Impacts of Economic Progress, Economic Structure, Urbanization Process, and Number of Vehicles on PM2.5Concentration: A Provincial Panel Data Model Analysis of China. Int. J. Environ. Res. Public Health 201916, 2926.
  • Singh, N.; Nyuur, R.; Richmond, B. Renewable Energy Development as a Driver of Economic Growth: Evidence from Multivariate Panel Data Analysis. Sustainability201911, 2418.
  1. Li, W.; Qi, X.; Zhao, X. Impact of Population Aging on Carbon Emission in China: A Panel Data Analysis. Sustainability201810, 2458.
  2. Zhao, H.; Guo, S.; Zhao, H. Impacts of GDP, Fossil Fuel Energy Consumption, Energy Consumption Intensity, and Economic Structure on SO2Emissions: A Multi-Variate Panel Data Model Analysis on Selected Chinese Provinces. Sustainability 201810, 657.
  3. Wang, J.; Li, L.; Zhang, F.; Xu, Q. Carbon Emissions Abatement Cost in China: Provincial Panel Data Analysis. Sustainability20146, 2584-2600.
  • Jin, T.; Kim, J. Coal Consumption and Economic Growth: Panel Cointegration and Causality Evidence from OECD and Non-OECD Countries. Sustainability201810, 660.
  • Armeanu, D.Ş.; Vintilă, G.; Gherghina, Ş.C. Does Renewable Energy Drive Sustainable Economic Growth? Multivariate Panel Data Evidence for EU-28 Countries. Energies201710, 381. Choi, Y.J.; Baek, J. Does FDI Really Matter to Economic Growth in India? Economies 20175, 20.
  1. Zhao, H.; Zhao, H.; Han, X.; He, Z.; Guo, S. Economic Growth, Electricity Consumption, Labor Force and Capital Input: A More Comprehensive Analysis on North China Using Panel Data. Energies20169, 891.

  1. Tao, R.; Glonț, O.R.; Li, Z.-Z.; Lobonț, O.R.; Guzun, A.A. New Evidence for Romania Regarding Dynamic Causality between Military Expenditure and Sustainable Economic Growth. Sustainability202012, 5053.
  2. Hatmanu, M.; Cautisanu, C.; Ifrim, M. The Impact of Interest Rate, Exchange Rate and European Business Climate on Economic Growth in Romania: An ARDL Approach with Structural Breaks. Sustainability202012, 2798.
  • Burciu, A.; Kicsi, R.; Bostan, I.; Condratov, I.; Hapenciuc, C.V. Sustainable Economic Growth Based on R&D Amplification and Technological Content of Exports. Evidences from Romania and The V4 Economies. Sustainability202012, 1831.
  • Andrei, J.; Mieila, M.; Popescu, G.H.; Nica, E.; Cristina, M. The Impact and Determinants of Environmental Taxation on Economic Growth Communities in Romania. Energies20169, 902.
  • Bucur, A.; Dobrotă, G.; Dumitraşcu, O. Implications of Fiscal Pressure on the Sustainability of the Equilibrium and Performance of Companies. Evidences in the Rubber and Plastic Industry from Romania. Sustainability201911, 2082.

  1. Dale, A.T.; Pereira de Lucena, A.F.; Marriott, J.; Borba, B.S.M.C.; Schaeffer, R.; Bilec, M.M. Modeling Future Life-Cycle Greenhouse Gas Emissions and Environmental Impacts of Electricity Supplies in Brazil. Energies20136, 3182-3208.

Author Response

Reply to Reviewer 2

Dear Colleague,

We are extremely grateful for your availability in reviewing this article, which has been resubmitted in revised version for publication. Your comments were very helpful in substantially improving the quality of the paper. Please find below the answers to your comments in a point-by-point manner.

1. It is good if you write the abstract again. You can view abstracts written in articles published in any MDPI journal.

Reply: Thank you very much for your observation. We have rewritten the abstract. Please find this change on page 1 of the revised version.  

Abstract. The study investigates through a panel data analysis the impact of factors such as non-performing loans, CO2 emissions, bank credit, inflation on the variable sustainable economic growth for India, Brazil and Romania during the period 2005–2017. Specifically, we investigate the timeline before, during and after the economic turmoil, with a special focus on the global financial crisis. Our empirical results are valuable for both developing and developed nations. As a first result, we showed that CO2 emissions increased the level of economic growth, but in this context authorities should design suitable policies to limit its impact on the overall society. In addition, the single supervision mechanism increased the level of sustainable economic growth. Last but not least, the period during and after the global financial crisis, sustainable economic growth decreased under the influence of bank credit, inflation and non-performing loans. Within this framework, public authorities are called to design efficient economic, fiscal and monetary policies.

  1. Keywords should be changed, for example: CO2 emissions, panel data analysis, GDP, Sustainability of Economic Growth, global financial crisis, credit, inflation, banking supervisory mechanisms.

Reply: Thank you for your suggestion. Following your advice, we have indicated the following keywords: CO2 emissions; gross domestic product; sustainable economic growth; single supervision mechanism; global financial crisis; inflation; bank credit; non-performing loans. 

3. “Banks play an important role in providing credit for green investment to mitigate CO2 emissions and achieve the sustainability of economic growth. As a result of increased credit, money supply multiplies, which in turn adds to inflation and any default in repayment results in non-performance of assets (known as non-performance of loans or NPL). On the other hand, higher investment leads to economic growth and it results in higher growth rate of the gross domestic product (GDP).” (76-81) and “Financing green investment is achieved with help from banking institutions. Therefore, bank loans (i.e., credit) are increasing and the money supply multiplies, which leads to inflation on the one hand and the growth of the GDP on the other hand.” (160-162) In my opinion, this should not be peculiar to loans only for green investments. It does not cause inflation, it acts as one of reasons for inflation. After all, the volume of green investments is not so much.

Reply: Thank you very much for your observation. We have rewritten the introduction and literature review sections and included relevant references. Please find these changes on pages 2-5 in the revised manuscript.   

4. “Brazil had achieved higher economy growth rates during the period 2003-2010, but this drastically has declined from 2011 onwards due to the impact of the global financial crisis”.(57-59) The world financial crisis of 2008-2009 more precise.

Reply: We have amended this, as indicated in our third reply.

5. “To the best of our knowledge, this appears to be the first research paper on this topic and we firmly believe that our findings will encourage further research in this direction”. (115-116)“…we firmly believe that our” change.

Reply: Thank you for your observation. We have amended the English wording, as follows: “To the best of our knowledge, these analyses mark the onset of studies on this topic, hence our findings might stir the interest of other inquiries in this particular direction.” Please find the change on page 3 of the revised manuscript. 

6. “According to specialists, the ineffective coordination of the two deepened the most recent global financial crisis”. (126-127) Who are these specialists?

Reply: We have deleted this wording from the revised version.

7. “Green economy promotes economic growth, which is inclusive and environmentally sustainable” (155-156) “In order to achieve the sustainability of economic growth, green investment is essenal” (159-160) Make adjustments.

Reply: Thank you for the observation. We have amended the typo.

8. The first limitation is the non-availability of data regarding CO2 emissions and non-performing loans for the period 2015–2017. (435-437) In this case, How was the study conducted?

Reply: We have reworded the limitation as following: “The first limitation is the non-availability of data regarding CO2 emissions and non-performing loans for the period 2015–2017, which however did not significantly influence our results”. Please find the change on page 15 of the revised manuscript. 

9. From the literature I mentioned below, in the “Literature review”, “Discussion”, “Conclusion and policy implications”. Use.

Reply: Thank you very much for your availability and kindness in providing such an extended list of sources. We are extremely grateful. These references are incorporated in the current version.

10. The article is good. The mentioned comments are quickly eliminated by specialists like you. I think the opinions of my and other reviewers are taken into account. Good luck to you.

Reply: Thank you very much for your kind remark and encouragement.

Reviewer 3 Report

Dear Authors,

Please find attached my kind Comments and Suggestions for you.

I wish you all the best and good luck!

Kind regards,

The Reviewer

Author Response

Reply to Reviewer 3

Dear Colleague,

We are extremely grateful for your availability in reviewing this article, which has been resubmitted in revised version for publication. Your positive comments were very helpful in substantially improving the quality of the paper. Please find below the answers to your comments in a point-by-point manner.

1) Suggestions meant to improve your current manuscript:

Dear Authors, I would kindly suggest making a correlation between investments and financial markets’ drastic fluctuations in developing the concept of social responsibility. I took the opportunity, in this regard, to make below a bibliographical suggestion, as a starting point: An Exploratory Study Based on a Questionnaire Concerning Green and Sustainable Finance, Corporate Social Responsibility, and Performance: Evidence from the Romanian Business Environment. J. Risk Financial Manag. 2019, 12, 162. DOI: 10.3390/jrfm12040162, link: https://www.mdpi.com/1911-8074/12/4/162 and Energy-Saving Potential of Applying Prefabricated Straw Bale Construction (PSBC) in Domestic Buildings in Northern China. Sustainability 2020, 12, 3464, https://doi.org/10.3390/su12083464.

Reply: Thank you for your observation and helpful suggestion. We have briefly mentioned the concept of social responsibility in the last section of our paper, since it is beyond the scope of our paper. We have referred and incorporated the expert views from the second article you recommended, along with other sources. Please find this change on page 15 of the revised manuscript. 

  1. Dear Authors, congratulations once again for your work and valuable insights reflected in the content of the manuscript, and I hope my comments will be of value to you! In addition, I also wish you good luck both with your current work as well as with your future projects!

Reply: We are extremely grateful for your encouragement and appreciation. Thank you very much.  

Reviewer 4 Report

  1. The authors analyzed the Sustainability of Economic Growth. This topic is very important for all stakeholders, both government, financial institutions, enterprises, and households.

  1. The title, although interesting, however, its performance raises some doubts. The subject is very extensive, and the study is rather cursory. The question arises whether in this analysis such serious conclusions can be drawn in the conclusion. The title is broad. After the focus is revised, the tile could be more specific.

  1. The authors made an unusual combination of countries: India, Brazil, and Romania. The question arises about the reasons for choosing the countries for the survey. What are the common parts of these countries, and what makes them different? The choice of countries should not be accidental.

  1. A literature review requires significant deepening. One has the impression that the Author did not reach into the basic items from the studied range.

  1. If the research method used in the work is applied only to the examined countries, then can the applications apply to the examined countries, or can they be applied.

  1. On p. 2, the authors write: "The sustainability of economic growth is a common goal for these three countries." I think this sentence applies not only to the three countries studied. Is it worth quoting them then? Is this not a truism? themselves in lone 422 they write: "We firmly believe that the sustainability of economic growth is the future of every economy"

  1. The research method is very narrowly described. The methodology should be broadly defined. I propose to include the subject of the research, the subject of the research, scope of research, examined indicators, the reference to sources, and an indication of research problems.

  1. On page 3, the authors present four research questions:

"The research questions of the present study are stated below: Did the global financial crisis affect the sustainability of economic growth? Did the banking supervisory mechanism influence the sustainability of economic growth? Did the global financial crisis and the banking supervisory mechanism jointly influence the sustainability of economic growth? What determinants of economic growth cause the sustainability of economic growth? "

Research questions are very extensive. I suggest you consider using one research question, but in more detail. These four questions give the possibility of analysis in at least four separate scientific articles. They are important and relevant, and their cursory presentation significantly impoverishes the analysis. The paper can be more focused.

  1. Abstract. After the focus is revised, the abstract could include more unique contributions and interesting findings.

  1. Introduction. Need to convince readers more about why the topic is important and for whom?

  1. Inline 405, the authors write: "Scientific literature reports numerous studies on economic growth." However, they do not give reference to literature.

  1. Is a reference to literature theory possible?

  1. The authors have to show what has been researched in the studied area, what research in the world has been made real, what new they have brought to the theory of literature, and what is new in the research from this article. What is unacceptable in this article, which is nowhere else.

  1. Technical notes, but important:

- It is necessary to include in the References all references in the text. E.g. The reference to the "2014 World Values ​​Survey" used in the introduction is not in the references.

- In the text, the authors refer to empirical data without including sources. For example, in the introduction, the description of GDP in Brazil does not refer to the source.

- line 422 "We firmly believe that the sustainability of economic growth is the future of every economy [8-9]" Why do the authors use a different record of the cited source [8-9]?

Author Response

Reply to Reviewer 4

Dear Colleague,

We are extremely grateful for your availability in reviewing this article, which has been resubmitted in revised version for publication. Your comments were very helpful in substantially improving the quality of the paper. Please find below the answers to your comments in a point-by-point manner.

1. The authors analyzed the Sustainability of Economic Growth. This topic is very important for all stakeholders, both government, financial institutions, enterprises, and households.

Reply:  We are indeed grateful for your compliments. We are extremely motivated to contribute in the future by tackling this line of research.

2. The title, although interesting, however, its performance raises some doubts. The subject is very extensive, and the study is rather cursory. The question arises whether in this analysis such serious conclusions can be drawn in the conclusion. The title is broad. After the focus is revised, the tile could be more specific.

Reply: Thank you for your observation. The amended title is “A Panel Data Analysis on Sustainable Economic Growth in India, Brazil and Romania”. We deem that the current title is adequate to emphasize the results of our study. 

3. The authors made an unusual combination of countries: India, Brazil, and Romania. The question arises about the reasons for choosing the countries for the survey. What are the common parts of these countries, and what makes them different? The choice of countries should not be accidental.

Reply: We have selected these nations with a purpose, which was explained as following: “The countries considered experience different levels of economic development, political instability, rule of law, social injustice, economic disparity and human development. Moreover, since they depend on agriculture and natural resources, these countries are also fast-growing economies. India, for instance, has the second largest population in the world and Brazil has the largest area of land, both countries being members of the BRICS group (i.e., acronym for the country association comprising Brazil, Russia, India, China, South Africa). Romania is a fast-growing economy among the newly admitted members of the European Union (EU). India and Brazil follow single supervisory mechanisms for the banking sector, while Romania follows a dual supervisory mechanism”.

Please find these changes on pages 1-2 of the revised manuscript.

4. A literature review requires significant deepening. One has the impression that the Author did not reach into the basic items from the studied range.

Reply: Thank you for your observation. We have thoroughly analyzed the specific literature and extended the reference list with relevant sources, included throughout the revised article. 

5. If the research method used in the work is applied only to the examined countries, then can the applications apply to the examined countries, or can they be applied.

Reply: The results of our research can be extended to other nations on the following rationale: a) We have selected nations from three different geographical regions: Romania from Europe, Brazil from South America and India from Asia; b) All three countries can be classified as emerging economies; c) Romania benefits from more than one supervision mechanism, while Brazil and India have a single supervisor mechanism; d) Romania has the highest GDP per capita and lowest population among these countries; India is the second largest populated country and it is the fifth largest economy of the world; Brazil has the second largest forest areas of the world. Therefore, these nations have different characteristics, which may be found in other nations of the world.

6. On p. 2, the authors write: "The sustainability of economic growth is a common goal for these three countries." I think this sentence applies not only to the three countries studied. Is it worth quoting them then? Is this not a truism? themselves in lone 422 they write: "We firmly believe that the sustainability of economic growth is the future of every economy".

Reply: Thank you for your observation. The wording has been removed from the revised manuscript.

7. The research method is very narrowly described. The methodology should be broadly defined. I propose to include the subject of the research, the subject of the research, scope of research, examined indicators, the reference to sources, and an indication of research problems.

Reply: Thank you for your observations. Following your valuable advice, we have extended section 3 containing the description of the research method. We have included details regarding the examined variables, data sources. Please find these changes on pages 5-6 of the revised manuscript. Details regarding the research question and study aims have been included in the introductory section. Please find such information on pages 1-4 of the revised manuscript.   

8. On page 3, the authors present four research questions:

"The research questions of the present study are stated below: Did the global financial crisis affect the sustainability of economic growth? Did the banking supervisory mechanism influence the sustainability of economic growth? Did the global financial crisis and the banking supervisory mechanism jointly influence the sustainability of economic growth? What determinants of economic growth cause the sustainability of economic growth? "

Research questions are very extensive. I suggest you consider using one research question, but in more detail. These four questions give the possibility of analysis in at least four separate scientific articles. They are important and relevant, and their cursory presentation significantly impoverishes the analysis. The paper can be more focused.

Reply: Thank you for your valuable suggestion. We have amended this aspect and formulated one meaningful research question, as follows: “The research question of the present study revolves around the investigation of sustainable economic growth in the context of the global financial crisis and the single supervision mechanism. More specifically, we analyze various factors that impacted sustainable economic growth in the aforementioned frameworks”. Please find these changes on page 3 of the revised manuscript.

9. Abstract. After the focus is revised, the abstract could include more unique contributions and interesting findings.

Reply: Thank you for your observation. We have revised the abstract of the article and included our main results.

10. Introduction. Need to convince readers more about why the topic is important and for whom?

Reply: The introduction have been thoroughly modified to include your suggestion. Please find these changes on pages 1-4 of the revised manuscript.

11. Inline 405, the authors write: "Scientific literature reports numerous studies on economic growth." However, they do not give reference to literature.

Reply: Thank you for your observation. We have amended this aspect and included sources such as Leitão (2010) or OECD (2014).

12. Is a reference to literature theory possible?

Reply: We have greatly extended the reference list in order to accommodate your suggestion by including both theoretical studies and practical analyses. This can be seen across the revised manuscript.  

13.The authors have to show what has been researched in the studied area, what research in the world has been made real, what new they have brought to the theory of literature, and what is new in the research from this article. What is unacceptable in this article, which is nowhere else.

Reply: Thank you very much for your valuable suggestions. We have detailed previous findings and the novelty of our research in sections such as “Introduction”, “Literature Review”, “Discussion” and “Conclusion and Policy Implications”. 

14. Technical notes, but important:

- It is necessary to include in the References all references in the text. E.g. The reference to the "2014 World Values ​​Survey" used in the introduction is not in the references.

- In the text, the authors refer to empirical data without including sources. For example, in the introduction, the description of GDP in Brazil does not refer to the source.

- line 422 "We firmly believe that the sustainability of economic growth is the future of every economy [8-9]" Why do the authors use a different record of the cited source [8-9]?

Reply: Thank you very much for the detailed suggestions. We have included the World Values Survey (2014) in the reference list. In terms of details regarding the Brazilian data, the source was indicated in the method section, i.e., World Development Indicators. Regarding your last observation, we have deleted this and rephrased the wording.

Round 2

Reviewer 1 Report

Major notes have been corrected.

Accept in present form.

Reviewer 3 Report

Dear Authors,

Good luck with your work!

Best regards,

The Reviewer

Reviewer 4 Report

Dear Authors,

As described in the Authors' responses, the text has been corrected. The problem is that the new text does not show the places that have been changed. I suggest that the Authors next time put the text in the review mode so that the changed places are more visible.
I wish you success in further work.